# Assembly of the Multi-Subunit Cytochrome *bc*_1_ Complex in the Yeast *Saccharomyces cerevisiae*

**DOI:** 10.3390/ijms231810537

**Published:** 2022-09-11

**Authors:** Vincenzo Zara, Gabriella De Blasi, Alessandra Ferramosca

**Affiliations:** Department of Biological and Environmental Sciences and Technologies, University of Salento, I-73100 Lecce, Italy

**Keywords:** mitochondria, respiratory chain, respiratory complexes, respiratory sub-complexes, respiratory super-complexes, complex III, mitochondrial biogenesis, chaperones, yeast strains

## Abstract

The cytochrome *bc*_1_ complex is an essential component of the mitochondrial respiratory chain of the yeast *Saccharomyces cerevisiae.* It is composed of ten protein subunits, three of them playing an important role in electron transfer and proton pumping across the inner mitochondrial membrane. Cytochrome *b*, the central component of this respiratory complex, is encoded by the mitochondrial genome, whereas all the other subunits are of nuclear origin. The assembly of all these subunits into the mature and functional cytochrome *bc*_1_ complex is therefore a complicated process which requires the participation of several chaperone proteins. It has been found that the assembly process of the mitochondrial *bc*_1_ complex proceeds through the formation of distinct sub-complexes in an ordered sequence. Most of these sub-complexes have been thoroughly characterized, and their molecular compositions have also been defined. This study critically analyses the results obtained so far and highlights new possible areas of investigation.

## 1. Introduction

The *bc*_1_ complex or complex III is a component of the respiratory chain of mitochondria and is responsible for the transfer of reducing equivalents from quinol to cytochrome *c* [1,2,3] (Figure 1). During this electron transfer, proton pumping from the matrix towards the intermembrane space also occurs, thereby generating an electrochemical potential across the inner mitochondrial membrane, which is necessary for ATP synthesis mediated by ATP synthase. 

The mitochondrial *bc*_1_ complex has been characterized in detail in yeast and in many other organisms, both structurally and functionally. In the yeast *S. cerevisiae*, this mitochondrial respiratory complex consists of ten protein subunits of different sizes (Table 1). 

Three subunits, cytochrome *b*, cytochrome *c*_1_ and the Rieske iron–sulfur protein (ISP), have a catalytic function in electron transfer and proton pumping. On the contrary, the remaining seven subunits, Qcr6p, Qcr7p, Qcr8p, Qcr9p, Qcr10p and the core 1 and core 2 proteins, have no definite roles in complex III activity. For this reason, they are also called supernumerary or non-catalytic subunits [1,2]. They are indeed absent in complex III of bacterial equivalents, such as *Paracoccus denitrificans* [4]. Little is known about the role of these subunits, although they are most probably involved in the process of assembly of this multi-subunit complex. It can also be hypothesized that they play a role in keeping complex III functionally active in yeast mitochondria. On the other hand, in mammalian mitochondria, the *bc*_1_ complex contains an additional supernumerary subunit which corresponds to the mitochondrial targeting sequence of ISP. This pre-sequence is incorporated into complex III after proteolytic processing [5]. The mature *bc*_1_ complex is a functional homodimer, firmly inserted in the inner membrane of yeast mitochondria where it interacts with other components of the respiratory chain such as the cytochrome *c* oxidase (complex IV), thus forming a super-complex between complex III and complex IV [6,7,8]. Larger associations between the respiratory complexes have been identified in mammalian mitochondria forming the so-called respirasomes [9,10,11].

Cytochrome *b* is the only protein subunit of the *bc*_1_ complex which is encoded by mitochondrial DNA [12], while all other subunits are encoded by the nuclear genome. This means that the remaining nine subunits of the yeast *bc*_1_ complex are firstly synthesized in the cytosol and are then imported into mitochondria thanks to specialized protein import machinery [13,14]. After import into mitochondria, a laborious process of multi-subunit assembly occurs and requires the participation of dedicated chaperones. The latter facilitate the correct folding and proper interaction of each protein subunit during *bc*_1_ complex biogenesis [3,6,15,16,17,18,19,20,21,22,23]. 

Several studies have been carried out on *bc*_1_ complex biogenesis in the yeast *S. cerevisiae*, and this has led to the proposal of a detailed model of assembly showing the progressive interaction of smaller *bc*_1_ sub-complexes during the maturation of this respiratory complex. However, some aspects of this complicated assembly process still need to be clarified. The aim of this review is to critically analyse the work carried out so far on this topic and to highlight new possible areas of investigation.

## 2. Structure of the Yeast Mitochondrial *bc*_1_ Complex 

Figure 2 illustrates the structure and the organization of the homodimeric *bc*_1_ complex in the inner mitochondrial membrane based on the crystal structure analysis carried out in yeast mitochondria [24,25,26] as well as in other organisms [27,28,29,30].

The central component of each homodimeric *bc*_1_ complex is cytochrome *b* [12], which is firmly inserted in the inner mitochondrial membrane by eight hydrophobic α-helices [24]. This protein is made up of 385 amino acids in the yeast *S. cerevisiae* and shows an apparent molecular mass of about 32 kDa when analysed on SDS–PAGE. Cytochrome *b* contains two prosthetic groups, heme *b*_L_ and heme *b*_H_, both involved in catalytic function and located in different portions of the protein [31]. Indeed, heme *b*_L_ is oriented towards the cytoplasmic surface, whereas heme *b*_H_ is positioned on the opposite site, towards the mitochondrial matrix. Cytochrome *c*_1_ [32], another catalytic subunit of the *bc*_1_ complex, interacts with cytochrome *b* and crosses the inner mitochondrial membrane with a single α-helix. It also contains a heme *c*_1_ prosthetic group, which is present in the catalytic domain of the protein exposed in the intermembrane space. The mature protein is composed of 248 amino acids (molecular mass of about 29 kDa on SDS–PAGE) but is preceded by a cleavable pre-sequence of 61 amino acids. The last catalytic subunit is the ISP [33], which shows a particular topology in the membrane because it possesses an α-helix located in one *bc*_1_ monomer, whereas the catalytic centre is present in the other monomer. The ISP is also characterized by the presence of a 2Fe–2S cluster, which is localized in the catalytic domain protruding in the intermembrane space. This catalytic protein contains 185 amino acids in its mature form (molecular mass of about 22 kDa on SDS–PAGE), which originates after the proteolytic cleavage of an amino-terminal pre-sequence of 30 amino acids.

The remaining seven non-catalytic subunits have different positions in the homodimeric *bc*_1_ complex. The two large core proteins, core protein 1 (431 amino acids) and core protein 2 (352 amino acids) [34,35], are located in the mitochondrial matrix, where they interact with the hydrophobic core of the *bc*_1_ complex, stably embedded in the inner mitochondrial membrane. Core proteins 1 and 2 show apparent molecular mass values of about 44 and 40 kDa when analysed on SDS–PAGE, respectively. In more detail, core protein 1, the largest subunit of the *bc*_1_ complex, is bound to the inner mitochondrial membrane but is mainly present in the mitochondrial matrix where it interacts with the smaller core protein 2. Qcr6p contains 122 amino acids and shows an apparent molecular mass of about 17 kDa on SDS–PAGE [36,37]. This protein is located in the intermembrane space where it interacts with the catalytic portion of cytochrome *c*_1_. Qcr7p, made up of 126 amino acids and showing a molecular mass of about 14 kDa [38,39], is present at the interface between the inner membrane and the matrix and interacts with the membrane-embedded cytochrome *b*. The smaller Qcr8p, made up of 93 amino acids and with a molecular mass of about 11 kDa [40], crosses the inner mitochondrial membrane with its single α-helix and is connected to cytochrome *b*. Similarly, Qcr9p [41,42], the smallest subunit of the *bc*_1_ complex made up of 65 amino acids (7.3 kDa), possesses a single transmembrane α-helix, which is embedded in the hydrophobic core of this respiratory chain complex. The crystal structure of the yeast *bc*_1_ complex [24] did not reveal the position of the Qcr10p, a subunit of about 8.5 kDa made up of 76 amino acids [43]. Nevertheless, the homologous protein in the bovine *bc*_1_ complex is characterized by a single α-helix which crosses the inner mitochondrial membrane in the vicinity of Qcr9p and of the ISP [28]. The membrane-spanning portion of the cytochrome *bc*_1_ complex, corresponding to the hydrophobic core of this respiratory complex, consists of 12 α-helices, 8 of which are provided by cytochrome *b*, and the remaining four deriving from cytochrome *c*_1_, ISP, Qcr8p and Qcr9p. Five out of the eight α-helices of cytochrome *b* are located at the dimer interface, whereas the remaining three are located in the periphery where they interact with the other *bc*_1_ subunits crossing the inner mitochondrial membrane [24,44]. Recent structural analysis of the yeast cytochrome *bc*_1_ revealed the presence of 12 molecules of phospholipids per homodimeric complex III, thereby suggesting the importance of these lipids in the structural arrangement of this respiratory complex [45].

The molecular mass of the homodimeric *bc*_1_ complex, analysed by the non-denaturating electrophoretic technique of Blue Native (BN)–PAGE, corresponded to about 670 kDa [8,46]. In native conditions, it was also possible to identify two respiratory super-complexes of higher molecular masses of about 850 and 1000 kDa [8,46]. The 850 kDa super-complex most probably was due to the interaction between the homodimeric *bc*_1_ complex and a monomer of the cytochrome *c* oxidase, whereas the 1000 kDa one contained the homodimeric *bc*_1_ complex bound to the homodimeric cytochrome *c* oxidase. It has been proposed that these respiratory super-complexes play a physiological role in electron transport by increasing the efficiency of the mitochondrial respiratory chain [9,47,48,49]. Nevertheless, some immature or incomplete forms of respiratory complexes and super-complexes appeared in various experimental conditions, thus stimulating different hypotheses on the functional interaction of these protein subunits and on their significance during mitochondrial oxidative phosphorylation. 

## 3. Assembly of the *bc*_1_ Complex Subunits

The assembly of the *bc*_1_ complex is a multi-step process and has been investigated in detail in *S. cerevisiae* mitochondria by several authors. In the past, the existence of various *bc*_1_ sub-complexes was hypothesized, and a tentative model of assembly was proposed on the basis of the data available at that time [50,51,52]. However, a significant step forward was made some years later thanks to a systematic analysis of several yeast mutant strains in which one or more genes encoding the *bc*_1_ subunits had been deleted. In this way, the role of each subunit during the assembly of the entire respiratory complex has been thoroughly investigated. 

Preliminary results were obtained with the SDS–PAGE analysis of the *bc*_1_ subunit composition in mitochondria isolated from various mutant yeast strains [53]. In these experimental conditions it was possible to investigate which subunits were firmly inserted in the mitochondrial membranes, and therefore protected from proteolytic degradation, when other subunits were absent [53]. Furthermore, these experiments suggested the existence of some *bc*_1_ assembly intermediates [53], which in part corresponded to those identified in previous studies [50,51,52]. However, this kind of analysis did not allow for the direct identification of hypothetical *bc*_1_ assembly intermediates because of the presence of the denaturing agent SDS. 

More studies were therefore carried out in non-denaturating conditions by extracting the mitochondrial proteins with the mild detergent digitonin and analysing them by BN–PAGE in the absence of SDS. In native conditions, distinct assembly intermediates or sub-complexes of the *bc*_1_ complex were clearly identified [8,46]. The single subunits constituting these assembly intermediates were then identified in the second dimension, carried out by SDS–PAGE, and subsequent immunodecoration with mono-specific antibodies [8,46]. Later, a different experimental approach, consisting of the use of an immunoprecipitation assay carried out in the presence of single *bc*_1_ subunits tagged with TAP or FLAG epitopes, led to the identification of the same assembly intermediates, thereby supporting the view that they represent bona fide intermediates during the assembly of the yeast cytochrome *bc*_1_ complex [54]. 

Further studies were carried out in various experimental conditions, thereby enriching the data of the assembly steps during complex III maturation [3,20,22]. This body of evidence has led to the proposal of a specific pathway of *bc*_1_ assembly in yeast mitochondria during which distinct intermediates were clearly identified and analysed at the level of single subunits. In the following paragraphs, the progressive process of assembly of the distinct subunits of the *S. cerevisiae bc*_1_ complex is reported in detail. 

### 3.1. The Synthesis of Cytochrome b

As stated before, cytochrome *b*, the central and hydrophobic subunit of the *bc*_1_ complex, is encoded by mitochondrial DNA and synthesized inside mitochondria. The nascent polypeptide chain of cytochrome *b* is co-translationally inserted into the inner mitochondrial membrane with the assistance of two chaperone proteins, Cbp3p and Cbp6p [15,18,19,55], which are present at the yeast ribosome tunnel exit. In more detail, Cbp3p interacts with the mitochondrial ribosome in proximity to Mrpl4p (mitochondrial ribosomal protein L4), thereby stabilizing the newly synthesized cytochrome *b* [20]. Cbp6p, on the other hand, is a translational activator of cytochrome *b* mRNA [55]. It has been proposed that the two chaperone proteins, Cbp3p and Cbp6p, allow for the coupled synthesis and assembly of the mitochondrial cytochrome *b* [20]. In addition, the Cbp3p–Cbp6p complex coordinates the synthesis of cytochrome *b* with the assembly of the entire *bc*_1_ complex [22]. Indeed, it was found that the expression of the mitochondrially encoded cytochrome *b* is dependent on the efficiency of the *bc*_1_ complex assembly [22]. However, it was also reported that the regulatory role of Cbp3p and Cbp6p in cytochrome *b* synthesis was evident only in some yeast strains, and that the main role of these chaperone proteins was related to cytochrome *b* hemylation [56]. The newly synthesized cytochrome *b* is inserted with eight α-helices in the inner mitochondrial membrane, where it interacts with the Cbp3p–Cbp6p complex. At this stage, also named intermediate 0 of the *bc*_1_ complex assembly [3], cytochrome *b* does not contain any prosthetic group (Figure 3). The introduction of the first heme group into the *b*_L_ site of cytochrome *b* [57] induces the subsequent binding of another chaperone protein, Cbp4p [17], thereby leading to intermediate I [3]. Cbp4p crosses the inner mitochondrial membrane with a single α-helix, but it mainly protrudes with its polypeptide chain in the intermembrane space (Figure 3). The binding of the second heme group at the *b*_H_ site of cytochrome *b* thus occurs with the concomitant release of the Cbp3p–Cbp6p complex [3]. 

### 3.2. The Early Core of the bc_1_ Complex

The fully hemylated cytochrome *b*, with Cbp4p still bound, is now able to interact with the two supernumerary subunits represented by Qcr7p and Qcr8p, thus forming the intermediate II [3]. This intermediate most probably corresponds to the “early core” which is the central hydrophobic core of the *bc*_1_ complex containing cytochrome *b*, Qcr7p and Qcr8p (Figure 3). This *bc*_1_ assembly intermediate was isolated for the first time in a stable form in some yeast deletion strains in which the genes encoding core protein 1 or core protein 2 had been deleted [8]. The *bc*_1_ early core showed a molecular mass of about 230 kDa when analysed in native conditions by BN–PAGE. This molecular size was too high on the basis of the molecular masses of the single subunits represented by cytochrome *b*, Qcr7p and Qcr8p, thereby suggesting the possible presence of other bound components. Interestingly, the co-migration of Cox6p (a subunit of the respiratory complex IV) in the same molecular mass region was found in BN–PAGE experiments, thus indicating a possible interaction of this early *bc*_1_ assembly intermediate with some subunits of the cytochrome *c* oxidase complex [8]. However, this does not exclude the interaction of the *bc*_1_ early core with other protein components, most probably the assembly factor Cpb4p as proposed by Ndi et al. (2008) [3]. In separate studies [54], an association of this *bc*_1_ early core with both Cbp3p and Cbp4p was found, hence indicating the need for further investigation on the first steps connected to the synthesis of cytochrome *b*. 

### 3.3. The Late Core of the bc_1_ Complex 

The late core of the *bc*_1_ complex is the subsequent assembly intermediate, which has been characterized in complex III biogenesis [46] (Figure 3). Other authors described some of the assembly intermediates of the *bc*_1_ complex very similar to this *bc*_1_ late core and named them intermediate III [3] or intermediate IV [58]. The *bc*_1_ late core, showing a molecular mass of about 500 kDa when analysed on BN–PAGE, is not only stable but is also productive because it was possible to convert it into the mature *bc*_1_ complex under suitable experimental conditions [59]. The existence of this *bc*_1_ late core has been demonstrated in three different yeast mutant strains, subsequently described, in which individual genes of the *bc*_1_ complex had been deleted [46]. Very interestingly, in each of the three deletion strains there were subtle differences in the structural compositions of the *bc*_1_ late core, thus providing valuable information on the last steps preceding the maturation of the yeast mitochondrial complex III.

The existence of the *bc*_1_ late core was first demonstrated in a yeast strain in which the gene encoding the supernumerary subunit Qcr9p had been deleted (Δ*qcr9*). When analysed by BN–PAGE, this assembly intermediate (Figure 4a) showed a molecular mass of approximately 500 kDa, which was significantly smaller than that of the mature and homodimeric *bc*_1_ complex corresponding to 670 kDa [8]. 

This assembly intermediate of 500 kDa, when analysed in the second dimension by SDS–PAGE and immunodecoration, revealed the presence of core protein 1, core protein 2, cytochrome *c*_1_ and Qcr6p in addition to the *bc*_1_ early core (cytochrome *b*, Qcr7p, Qcr8p) [8]. Very interestingly, in the Δ*qcr9* strain analysed in the same experimental conditions, the ISP migrated as a single protein subunit of about 35 kDa. This suggests that Qcr9p is necessary for the binding of this last catalytic subunit to the *bc*_1_ late core. In addition, this assembly intermediate also contained the chaperone Bcs1p, which has been suggested to play a role in the binding of the ISP to the immature *bc*_1_ complex [6,16]. In all the experiments previously reported [8,46], the *bc*_1_ late core was identified by BN–PAGE in the presence of the mild detergent digitonin, which was therefore able to maintain the integrity of this 500 kDa sub-complex. Further experiments were also carried out in native conditions with the use of the detergent Triton X-100, which was also able to preserve the integrity of the *bc*_1_ late core [46]. Overall, in the *bc*_1_ late core identified and analysed in the Δ*qcr9* strain, three subunits are missing: Qcr9p, ISP and Qcr10p.

A second form of the *bc*_1_ late core (Figure 4b) was subsequently found in a yeast deletion strain missing the gene encoding the ISP (Δ*isp*). This assembly intermediate showed, when analysed in native conditions followed by SDS–PAGE and immunodecoration, a subunit composition quite similar to that of the previous *bc*_1_ late core (i.e., cytochrome *b*, QCr7p, Qcr8p, representing the early core, and core protein 1, core protein 2, cytochrome *c*_1_, Qcr6p), plus the supernumerary subunit Qcr9p. It is also interesting to underline that this *bc*_1_ assembly intermediate contained the chaperone protein Bcs1p, as already found in the previously reported late core intermediate. Overall, in this assembly intermediate, the missing subunits are ISP and Qcr10p. This suggests that (a) the binding of Qcr9p to this *bc*_1_ assembly intermediate is necessary for the subsequent binding of the ISP, and that (b) its binding precedes that of this last catalytic subunit. 

A third form of the *bc*_1_ late core (Figure 4c) was identified in a yeast mutant strain in which the gene encoding Bcs1p had been deleted (Δ*bcs1*). In this case, all the *bc*_1_ subunits were identified in the approximately 500 kDa intermediate except for Bcs1p, ISP and Qcr10p. This finding suggests that the presence of the chaperone protein Bcs1p is required for the insertion of the catalytic subunit ISP, to which the binding of the last supernumerary subunit Qcr10p follows. 

Finally, all the late core intermediates of the *bc*_1_ complex analysed in the three yeast deletion strains mentioned above showed no binding at all or only a loose association with the cytochrome *c* oxidase complex. This behaviour of the *bc*_1_ late core differed from that of the *bc*_1_ early core that was found in association with complex IV of the respiratory chain [8,46]. Notably, in all the yeast deletion strains analysed, Cox6p migrated in a molecular mass region of about 230 kDa, in which the *bc*_1_ early core was previously identified. In addition, all the yeast deletion strains (Δ*qcr9*, Δ*isp* and Δ*bcs1*) containing the *bc*_1_ late core were unable to respire on non-fermentable media [46]. We can therefore conclude that the absence of either of the two *bc*_1_ subunits, Qcr9p and ISP, or of the chaperone Bcs1p determines the appearance of an immature form of the *bc*_1_ complex with a molecular mass of about 500 kDa. It is also important to note that the experiments carried out with the yeast deletion strains Δ*qcr9*, Δ*isp* and Δ*bcs1* suggested the sequential binding of Qcr9p, ISP and Qcr10p to this late *bc*_1_ core intermediate [46]. 

As stated before, the *bc*_1_ late core was unable to stably associate with the cytochrome *c* oxidase complex to form a respiratory super-complex. On the contrary, the mature *bc*_1_ complex formed a stable super-complex in association with the cytochrome *c* oxidase complex. Interestingly, the N-terminal domain (92 residues) of the ISP was able to induce the stabilization of a super-complex consisting of the *bc*_1_ late core and the cytochrome *c* oxidase complex [60]. This stabilization effect was independent of the presence of Bcs1p, thus indicating that its chaperone-mediated translocation effect across the inner membrane is not required. It was further proposed that the stabilization effect exerted by the N-terminal domain of the ISP was due to an enhanced amount of the cytochrome *c* oxidase complex [60]. It therefore appears that super-complex stabilization occurs when there is an increased formation of complex IV in mitochondria.

The nature of the true assembly intermediate of this *bc*_1_ late core is also supported by its identification in further yeast deletion strains such as Δ*isp*/Δ*qcr6*, Δ*isp*/Δ*qcr9*, Δ*isp*/Δ*qcr10* and Δ*qcr9*/Δ*qcr10*. In all these double deletion strains, an assembly intermediate of about 500 kDa was repeatedly found [46]. Very interestingly, another yeast mutant strain in which both genes encoding Qcr6p and Qcr9p had been deleted (Δ*qcr6*/Δ*qcr9*) showed the same *bc*_1_ assembly intermediate and suggested that the supernumerary subunit Qcr6p was not necessary for its stability [46]. Furthermore, in this last yeast double deletion strain, the 500 kDa intermediate showed its minimal composition (Figure 4d) because only cytochrome *b*, Qcr7p, Qcr8p, core protein 1, core protein 2 and cytochrome *c*_1_ were found, whereas all the remaining subunits, Qcr6p, Qcr9p, ISP and Qcr10p, were missing. Furthermore, it must be considered that Bcs1p was still bound to this form of the *bc*_1_ late core, as was already found in the case of the yeast deletion strains Δ*isp* and Δ*qcr9*. Interestingly, the existence of this minimal form of the *bc*_1_ late core suggests that apart from the *bc*_1_ early core (cytochrome *b*, Qcr7p and Qcr8p), there is an early and stable association also among cytochrome *c*_1_ and the two core proteins 1 and 2. This finding will be discussed later since distinct sub-complexes between cytochrome *c*_1_ and each of the two core proteins were specifically isolated [46].

### 3.4. Dynamics of the Late Core of the bc_1_ Complex 

Subsequent experiments demonstrated that the 500 kDa assembly intermediate was not only stable in different experimental conditions [46] but was also productive based on its ability to restore a functional dimeric *bc*_1_ complex and subsequently the respiratory super-complexes [59]. To this end, the chaperone protein Bcs1p was expressed in a yeast deletion strain in which the gene encoding Bcs1p had been deleted (Δ*bcs1*) [59]. Indeed, in previous studies, an essential role of Bcs1p for the correct insertion of the ISP into an immature *bc*_1_ complex was proposed [6,16]. Bcs1p primarily translocates the ISP across the inner mitochondrial membrane into the matrix [61], where the insertion of the 2Fe–2S cluster occurs [62]. Then, the same chaperone retranslocates the C-terminal domain of the ISP with the FeS cluster bound towards the cytosolic surface of the inner mitochondrial membrane. In this respect, it is also important to note the role of Mzm1p (mitochondrial zinc maintenance protein 1), a further identified chaperone required for the *bc*_1_ complex assembly [63,64], which prevents misfolding of the ISP in the matrix [21]. A human ortholog of *MZM1*, named *LYRM7*, has been identified in humans, and a pathogenic mutation caused cytochrome *bc*_1_ complex functional defects [65,66].

The expression of *BCS1* in the Δ*bcs1* strain converted the 500 kDa assembly intermediate into the mature and functional *bc*_1_ complex with a molecular mass of about 670 kDa [59]. Interestingly, the molecular analysis of the recovered *bc*_1_ complex, induced by Bcs1p overexpression, revealed not only the insertion of the ISP but also the sequential addition of the supernumerary subunit Qcr10p. In addition, this recovered homodimeric *bc*_1_ complex was able to interact with the cytochrome *c* oxidase complex, thereby reconstructing the respiratory super-complexes of 850 and 1000 kDa. 

In humans, point mutations of the *BCS1L* ortholog were responsible for the appearance of several *bc*_1_ complex defects, characterized by different clinical manifestations [67,68,69,70,71,72]. Based on the sequence similarities between the yeast Bcs1p and the protein encoded by the human ortholog *BCS1L*, some mutated yeast Bcs1ps were expressed in the Δ*bcs1* strain [59]. The yeast mutant Bcs1ps contained distinct amino acid substitutions corresponding to those found in human Bcs1lp and responsible for the appearance of human pathologies. The aim of these experiments was to gain more information on the role of these amino acids, which were present in critical regions of the polypeptide chain of both yeast Bcs1p and human Bcs1lp [59]. Interestingly, arginine in position 81 was not essential for Bcs1p targeting into yeast mitochondria and therefore for its chaperone function. Indeed, the (R81C) yeast Bcs1p was able to recover the *bc*_1_ late core, promoting the integration of ISP and Qcr10p into this assembly intermediate. On the contrary, two other point mutations (K192P and F401I) in distinct domains of the yeast Bcs1p (Bcs1p-specific domain and AAA domain) were unable to recover the *bc*_1_ late core, hence confirming the essential roles of these amino acids for the chaperone activity of Bcs1p both in yeast and in humans [59].

On the other hand, with a different experimental approach, it was demonstrated that also the early core represented a productive intermediate during the *bc*_1_ complex assembly. In the Δ*isp*/Δ*cor2* yeast strain, analysed in native conditions followed by SDS–PAGE and immunodecoration, only the early core of about 230 kDa was detected because of the absence of core protein 2, which blocked the assembly of the mature *bc*_1_ complex. Very interestingly, the overexpression of Cor2-TAP in this yeast double deletion strain led to the appearance of a *bc*_1_ late core of about 500 kDa. Indeed, in these conditions, the only missing subunit was the ISP, which froze complex III at the level of the *bc*_1_ late core [54]. These results demonstrated the nature of the bona fide intermediate of the 230 kDa early core, which was able, under suitable experimental conditions, to progress towards a subsequent *bc*_1_ assembly intermediate of higher molecular mass. 

Due to the particular topology of the ISP in the *bc*_1_ complex, i.e., the catalytic centre in one monomer and the transmembrane helix in the other, it has been hypothesized that this protein may have a role in *bc*_1_ complex dimerization [2,46]. To investigate this possibility, a different experimental strategy was followed, consisting of the isolation of mitochondria from a yeast strain in which the gene encoding the ISP had been deleted (Δ*isp*) and in which the expression of Cor2-TAP had been induced. In these cells, as stated before, the absence of the ISP caused the incomplete assembly of the *bc*_1_ complex and the appearance of a *bc*_1_ late core of approximately 500 kDa. The immunoprecipitation assay of mitochondria isolated from these mutated cells in which the expression of Cor2-TAP had been induced allowed us to investigate the structural properties of this late core intermediate [54]. Very interestingly, the expression of Cor2-TAP in this yeast strain induced the incorporation of this modified protein into the *bc*_1_ late core together with native core protein 2, thus demonstrating that this intermediate was already in a dimeric form. 

Independent experiments carried out with the overexpressed Qcr9-FLAG in the same yeast deletion strain (Δ*isp*) led to the co-isolation of Qcr9p and Qcr9-FLAG, hence further confirming the dimeric structure of the *bc*_1_ late core. Altogether, these findings also demonstrate that the presence of the ISP is not necessary for *bc*_1_ complex dimerization, which happens before late core formation. Another interesting aspect revealed by these immunoprecipitation experiments is the interaction, even if incomplete and non-productive, between the *bc*_1_ late core and the cytochrome *c* oxidase complex [54]. Although the ISP is not necessary for *bc*_1_ complex dimerization, it is still possible that this catalytic subunit may participate in the formation of a stable and functionally active super-complex between complex III and complex IV. 

Very interestingly, the analysis of the dynamics of complex III assembly in a study directed towards the role played by Cbp3p and Cbp6p in early steps of cytochrome *b* biogenesis revealed the true nature of the distinct *bc*_1_ complex assembly intermediates found in several yeast mutant strains lacking each of the catalytic or supernumerary subunits [22]. This indicates the productive character of all these intermediates during the assembly of the mitochondrial *bc*_1_ complex. 

### 3.5. The Final Steps of Cytochrome bc_1_ Maturation 

The last steps of cytochrome *bc*_1_ assembly (Figure 3) involve the insertion of the ISP, the last catalytic subunit, and the final association with the supernumerary subunit Qcr10p [73]. 

After the binding of the last subunit Qcr10p to the ISP, the homodimeric *bc*_1_ complex is finally obtained. As previously reported, the homodimeric *bc*_1_ complex in yeast mitochondria is also able to associate with the cytochrome *c* oxidase complex, hence forming a respiratory super-complex. Under experimental conditions in which the mitochondrial membranes, instead of intact mitochondria, were used for the BN–PAGE analysis, the homodimeric *bc*_1_ complex alone (670 kDa) or bound to a monomer (850 kDa) or a dimer of cytochrome *c* oxidase (1000 kDa) was reproducibly found [8]. On the other hand, the use of intact mitochondria [7,47] instead of the isolated mitochondrial membranes [8,46] led to the isolation of the only band of 1000 kDa with minor amounts of the 850 kDa band. As discussed below, further studies are still necessary to identify when and how the interaction between these two respiratory complexes occurs in yeast mitochondria. 

## 4. Other Assembly Intermediates during *bc*_1_ Complex Maturation

Cytochrome *c*_1_ and the two core proteins 1 and 2 were the most resistant *bc*_1_ subunits when some yeast mutant strains, lacking Qcr7p and/or Qcr8p, were analysed [53]. When the mitochondrial membranes isolated from these mutant yeast strains were analysed by BN–PAGE followed by SDS–PAGE and immunodecoration, a sub-complex of about 100 kDa was identified containing cytochrome *c*_1_ bound to core protein 2 (Figure 5a). 

This quite unexpected finding was also confirmed in a mutant yeast strain in which the gene encoding the core protein 1 had been deleted [8]. On the contrary, in another yeast mutant strain in which the gene encoding the core protein 2 had been deleted, the appearance of a new sub-complex of about 78 kDa and made up of cytochrome *c*_1_ bound to core protein 1 was found (Figure 5b) [8]. Altogether, these data demonstrate the capability of these three proteins, cytochrome *c*_1_, core protein 1 and core protein 2, to interact each other, thus increasing their stability in the absence of an assembled form of the *bc*_1_ complex. Independent experiments, carried out in a yeast mutant strain in which the genes encoding both core proteins had been deleted (Δ*cor1*/Δ*cor2*) and in which Cor2-TAP was overexpressed, revealed the association of cytochrome *c*_1_ with Cor2-TAP [54]. This finding reinforced the previous results and suggested early and preferential binding of this catalytic subunit to the core protein 2 when the *bc*_1_ complex assembly is blocked at the level of the early core. The interaction between cytochrome *c*_1_ and the core proteins in an unassembled *bc*_1_ complex is quite unexpected, but it is possible that the N-terminal pre-sequence of this catalytic subunit interacts with the core proteins during the import into mitochondria. In fact, the two core proteins are relatives of the α and β subunits of the MAS-encoded matrix processing peptidase present inside mitochondria [74]. However, differently from plants, the yeast core proteins do not show any protease activity. It is also noteworthy to observe that in the crystal structure of the mature *bc*_1_ complex, cytochrome *c*_1_ is in the vicinity of core protein 1 but not of core protein 2 (Figure 5b vs. Figure 5a). However, the structural organization of the proteins can be different during the process of assembly of a multi-subunit complex in comparison to the mature form acquired after insertion into the inner mitochondrial membrane. 

A strong interaction between the two core subunits was repeatedly found during the experiments aimed at the identification of *bc*_1_ complex assembly intermediates [8,53]. Interestingly, the existence of a further assembly intermediate, named core 1/2 module, has been recently proposed [58]. It was also hypothesized that this heterotetrameric module was important for *bc*_1_ dimerization after its sequential addition to the so-called intermediate II, thereby forming the dimeric intermediate III [58]. However, further studies are required to clarify the molecular mechanism leading to *bc*_1_ complex dimerization, especially with respect to the possible role played by other subunits, such as the chaperone Cbp4p and the catalytic protein cytochrome *c*_1_, during this process. 

As previously reported, the ISP represents the last catalytic subunit inserted into the *bc*_1_ late core. In the absence of Bcs1p or Qcr9p, this catalytic subunit was found in a molecular mass region of about 35 kDa, thus indicating the lack of its insertion into the *bc*_1_ late core in the absence of either of the two subunits. Very interestingly, in the yeast mutant strains lacking core protein 1 or core protein 2, and in which the *bc*_1_ early core was previously identified, a sub-complex of about 66 kDa, containing the ISP associated with Qcr9p, was clearly found (Figure 5c). This sub-complex was also identified in a different yeast mutant strain in which the gene encoding cytochrome *c*_1_ had been deleted [8]. As previously reported, the ISP presents a trans-dimeric structure in the crystallographic studies carried out with the mature *bc*_1_ complex [24]. It is therefore possible that this catalytic subunit interacts with the Qcr9p, belonging to the other monomer in the sub-complex of 66 kDa previously identified.

## 5. Conclusions and Perspectives

The experiments carried out so far on the assembly pathway of the mitochondrial *bc*_1_ complex in *S. cerevisiae* have clearly established the formation of distinct sub-complexes during the maturation of this respiratory complex. Based on these data, it is reasonable to exclude the sequential and linear addition of single subunits to a growing complex III. Indeed, the results obtained in several experiments, carried out in various laboratories, have demonstrated the existence of stable sub-complexes during the *bc*_1_ complex assembly. In addition, these sub-complexes were also productive because they represented bona fide intermediates capable of regenerating the mature and functional *bc*_1_ complex under suitable experimental conditions. On the other hand, comparatively little is known about the regulatory mechanisms of the distinct steps identified during the *bc*_1_ complex assembly in yeast mitochondria. Some studies have been carried on the characteristics and the regulation of the initial steps of *bc*_1_ assembly in yeast mitochondria [3,20,22,56], although several aspects of this multi-step process await clarification. 

Overall, there is general agreement on the structure of the distinct sub-complexes identified during the yeast *bc*_1_ complex assembly. The first sub-complex found during the growth of the nascent *bc*_1_ complex is intermediate 0, constituted by the catalytic subunit cytochrome *b* bound to the Cbp3p–Cbp6p chaperones [3,57,58]. Then intermediate I follows, made up of the partially hemylated cytochrome *b* bound to Cbp3p–Cbp6p and Cbp4p [3,57,58]. The binding of the two supernumerary subunits Qcr7p and Qcr8p to the central component cytochrome *b,* with its simultaneous full hemylation, leads to the release of the hetero-dimer Cbp3p–Cbp6p but not of Cbp4p. This new assembly intermediate was named intermediate II [3,57,58] or *bc*_1_ early core [8,46]. An early and fundamental role of the two supernumerary subunits Qcr7p and Qcr8p has been proposed in the past [8,39,53,75], independently of the role played by the chaperone proteins Cbp3p, Cbp4p and Cbp6p. It is therefore essential to investigate further the very early steps leading to the first sub-complex identified, i.e., that containing the newly synthesized cytochrome *b* associated with some chaperone proteins and some supernumerary subunits. This also in view of the possible regulatory role of these first events, which coordinate the synthesis of the nuclear-encoded cytochrome *b* with that of the numerous subunits encoded by the nuclear genome. It will in fact be possible to identify further and unknown chaperone proteins involved in this process, as already suggested by some authors [56]. Interestingly, regulation of cytochrome *b* synthesis has been proposed by a feedback loop in which the Cbp3p–Cpb6p complex plays a role during the first steps of *bc*_1_ complex assembly [3,22]. Nevertheless, further studies are needed to better clarify this important point of regulation and coordination.

After the synthesis of intermediate II or the *bc*_1_ early core, there is a transition towards a larger sub-complex, identified as intermediate III (made up of cytochrome *b*, Qrc7p, Qcr8p, core proteins 1 and 2) and intermediate IV (made up of the previous subunits plus cytochrome *c*_1_ and Qcr6p) [58]. Interestingly, in our hands, we reproducibly found an assembly intermediate of about 500 kDa, most probably corresponding to intermediate IV [46]. Finally, the complete assembly of the cytochrome *bc*_1_ complex is followed by the sequential binding of the last subunits represented by Qcr9p, ISP and Qcr10p, respectively [46]. Both Qcr9p and Bcs1p are essential for the binding of the ISP to the *bc*_1_ late core. However, the presence of only one of these two proteins is not able to substitute the other in the binding of the ISP. Interestingly, the insertion of this last catalytic subunit does not remove Bcs1p from the mature *bc*_1_ complex that, indeed, was still found in the 670 kDa band, which corresponds to the homodimeric *bc*_1_ complex. 

Some lessons have been learned through yeast gene deletion experiments. The most dramatic effects were obtained in the absence of each of the two core proteins or of the supernumerary subunits Qcr7p and/or Qcr8p. In these conditions, the assembly of the yeast cytochrome *bc*_1_ complex was severely affected, thus giving rise to the appearance of a large set of small *bc*_1_ sub-complexes [8]. On the other hand, the absence of Qcr9p blocked the assembly of this respiratory complex at a later stage with the formation of the *bc*_1_ late core, which, although larger, was similarly non-functional [8,46]. On the contrary, the absence of the supernumerary subunits Qcr6p and Qcr10p apparently did not affect the assembly process of the yeast cytochrome *bc*_1_ complex. Indeed, Qcr6p was neither required for the interaction of the other *bc*_1_ subunits, and therefore for the assembly of the homodimeric *bc*_1_ complex, nor for the formation of the super-complex between complex III and complex IV. Based on the available data, the role of Qcr6p during the assembly pathway of the *bc*_1_ complex is still enigmatic. Previous studies carried out in a yeast strain lacking Qcr6p have revealed only a retardation in the cytochrome *c*_1_ maturation [53]. In this context, it is worth underlying that a functional interaction between cytochrome *c*_1_ and Qcr6p has been previously proposed [76]. Similarly, the absence of Qcr10p did not interfere with the incorporation of all the remaining *bc*_1_ subunits into a functional respiratory complex, which was also able to interact with the cytochrome *c* oxidase complex. Overall, all these experiments demonstrate the participation of almost all supernumerary subunits, even if to a different extent, in the assembly process of the *bc*_1_ complex. Nevertheless, it is still unclear whether these non-catalytic subunits play additional roles in this respiratory complex by contributing, for example, to the stability or dynamics of the complex.

Up until now, several assembly factors of the yeast mitochondrial *bc*_1_ complex have been described. They are represented by Cbp3p, Cbp4p, Cbp6p, Bcs1p and Mzm1p [15,16,17,55,63,64]. Although some hints on the role of these chaperone proteins during the assembly of the mitochondrial *bc*_1_ complex have been already obtained, further studies are necessary to clarify the details of the molecular mechanisms involved in their actions. In this context, very little is known about the role played by Bca1p, another assembly factor of the respiratory complex III, which appears to act upstream of the ISP insertion in the assembly process [77]. More information is available for the chaperone protein Bcs1p, which is clearly required for the binding of the ISP to the *bc*_1_ late core [46,59]. Bcs1p was found stably bound to the *bc*_1_ late core of about 500 kDa on BN–PAGE [46,59]. Interestingly, this chaperone protein was reproducibly found also in the homodimeric *bc*_1_ complex of about 670 kDa formed after the incorporation of the ISP. Therefore, the role played by Bcs1p deserves further investigation, especially regarding its presence in the mature and functional *bc*_1_ complex. 

A long-discussed topic regards how and when the *bc*_1_ complex dimerization occurs. The results so far obtained indicate that the dimerization is an early event in *bc*_1_ assembly [54], and this is understandable on the basis of the structural properties of the membrane-embedded cytochrome *b*. The hydrophobic helices of this catalytic subunit, in fact, interact with those of another cytochrome *b* in the lipid environment of the inner mitochondrial membrane independently of other catalytic or non-catalytic subunits. This suggests an early and rapid gluing between the cytochrome *b* subunits belonging to each *bc*_1_ monomer. Interestingly, subsequent experiments carried out on the timing of *bc*_1_ complex dimerization basically confirmed the early characteristics of homodimer formation [54] with the addition of further elements [58]. Indeed, it was found that dimerization occurred at the level of intermediate III containing cytochrome *b* bound to the small supernumerary subunits Qcr7p and Qcr8p and to the two core proteins 1 and 2 [58]. It was concluded that the presence of cytochrome *c*_1_ was dispensable for *bc*_1_ complex dimerization, and that the two core proteins 1 and 2 played a fundamental role in this respect. Furthermore, it was hypothesized that the release of Cbp4p from intermediate II and then the transition towards intermediate III (with the core proteins bound) induced *bc*_1_ complex dimerization. Very interestingly, both studies [54,58] excluded a role of the ISP in this process. 

Another aspect of interest is the interaction between the *bc*_1_ complex and the cytochrome *c* oxidase complex in the formation of the so-called super-complexes found in distinct experimental conditions. More extensive interactions between the respiratory complexes of the inner mitochondrial membranes have been found in mammals and other organisms [9,78,79,80,81]. Surprisingly, an interaction between the *bc*_1_-oxidase super-complex and the TIM23 machinery was also described [82]. In addition, the *bc*_1_-oxidase super-complex bound to the TIM23 machinery was found in association with the ADP/ATP carrier of the inner mitochondrial membrane [83]. These interactions reveal the complexity and the versatility of mitochondrial activities, suggesting that mitochondrial energetics and metabolism are closely integrated with protein biogenesis. It is also worth noting that the cytochrome *c* oxidase complex can interact with the *bc*_1_ early core, composed of cytochrome *b*/Qcr7p/Qcr8p (and most probably Cbp4p), but not in a stable manner, with the *bc*_1_ late core containing other *bc*_1_ subunits besides those of the early core. Further experiments are therefore necessary to clarify if the preliminary interaction between the *bc*_1_ early core and complex IV represents a true assembly intermediate during the biogenesis of these respiratory complexes or of it is an occasional and non-productive interaction. In any case, it was found that the assembly of the cytochrome *c* oxidase complex occurred independently of that of the *bc*_1_ complex, and therefore also when the formation of the *bc*_1_ early core was blocked [7,47,53]. Recent structural studies revealed that core protein 1, in addition to its interaction with core protein 2, is also involved in the interaction with the cytochrome *c* oxidase complex, hence contributing to the formation of the III–IV respiratory super-complex [45,84]. In addition, it has been found that cardiolipin also plays a role in facilitating the interaction between the mitochondrial complexes III and IV in the yeast *S. cerevisiae* [85,86,87]. However, this does not exclude the possibility that further membrane phospholipids may be involved in the interaction between these respiratory complexes. 

Undoubtedly, the studies carried out in *S. cerevisiae* are facilitated by the techniques of yeast genetics and by the possibility of growing the cells in different experimental conditions, hence modulating their metabolism and mitochondrial energy production. In other words, this makes it possible to use a more flexible experimental system which offers several advantages in comparison to more complex organisms. This also means that the yeast cytochrome *bc*_1_ complex can be used as a model to clarify the molecular mechanisms of multi-subunit protein assembly. This has led to the dissection of the cytochrome *bc*_1_ complex assembly in distinct steps, allowing the identification of several *bc*_1_ sub-complexes and proposing their orderly interaction during the maturation of this respiratory complex. Despite the considerable work done so far, further aspects concerning the molecular mechanisms of the assembly of complex III and its regulation require further and more in-depth studies. This may also be helpful in clarifying the pathogenesis of certain human mitochondrial diseases due to a malfunctioning of the cytochrome *bc*_1_ complex.

## Figures and Tables

**Figure 1 ijms-23-10537-f001:**
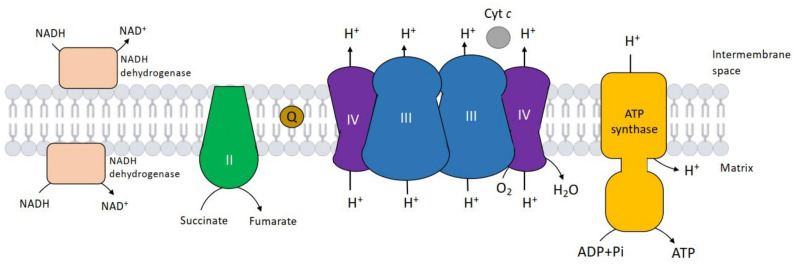
Location of the cytochrome *bc*_1_ complex (or complex III) in the inner mitochondrial membrane of *Saccharomyces cerevisiae* as part of the mitochondrial respiratory chain. Complex III is shown in association with complex IV (respiratory super-complex).

**Figure 2 ijms-23-10537-f002:**
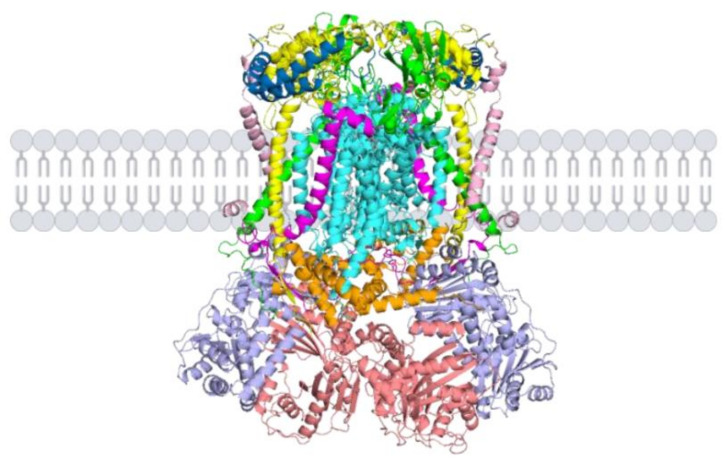
Structure of the homodimeric *bc*_1_ complex in the inner mitochondrial membrane based on the crystal structure analysis carried out in yeast mitochondria (cytochrome *b*: aquamarine, cytochrome *c*_1_: yellow, ISP: green, core protein 1: light blue, core protein 2: salmon, Qcr6p: sky blue, Qcr7p: orange, Qcr8p: pink, Qcr9p: magenta).

**Figure 3 ijms-23-10537-f003:**
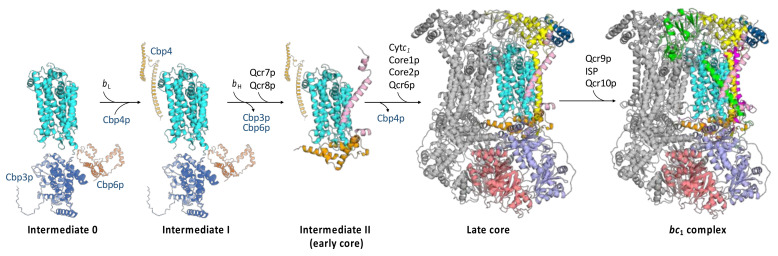
Schematic model depicting the putative multi-step assembly of the mitochondrial *bc*_1_ complex (cytochrome *b*: aquamarine, cytochrome *c*_1_ (Cyt*c*_1_): yellow, ISP: green, core protein 1 (Core1p): light blue, core protein 2 (Core2p): salmon, Qcr6p: sky blue, Qcr7p: orange, Qcr8p: pink, Qcr9p: magenta). Cbp3p, Cbp4p and Cbp6p structures were obtained from the UniProt database (https://www.uniprot.org) (accessed on 24 August 2022) using the protein structure prediction tool AlphaFold. The interaction sites between chaperone proteins and *bc*_1_ subunits remain unknown.

**Figure 4 ijms-23-10537-f004:**
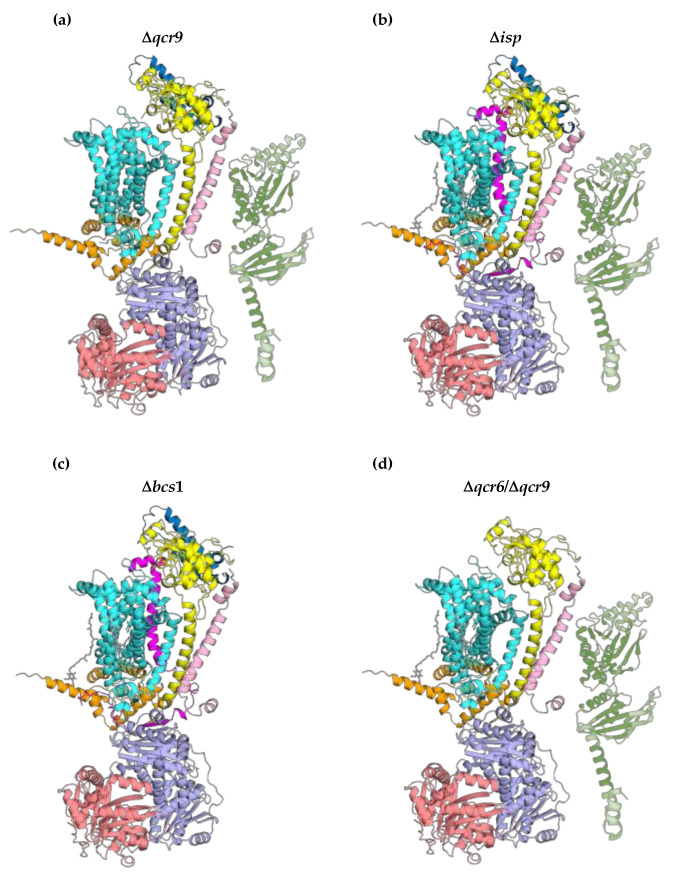
Different forms of the *bc*_1_ late core identified in four yeast mutant strains: (**a**) Δ*qcr9*; (**b**) Δ*isp*; (**c**) Δ*bcs1*; (**d**) Δ*qcr6*/Δ*qcr9.* The *bc*_1_ late core is depicted in its monomeric form (cytochrome *b*: aquamarine, cytochrome *c*_1_: yellow, core protein 1: light blue, core protein 2: salmon, Qcr6p: sky blue, Qcr7p: orange, Qcr8p: pink, Qcr9p: magenta). The Bcs1p structure was obtained from the UniProt database (https://www.uniprot.org) (accessed on 24 August 2022) using the protein structure prediction tool AlphaFold. The interaction sites between Bcs1p (protein coloured in green) and *bc*_1_ late core subunits remain unknown.

**Figure 5 ijms-23-10537-f005:**
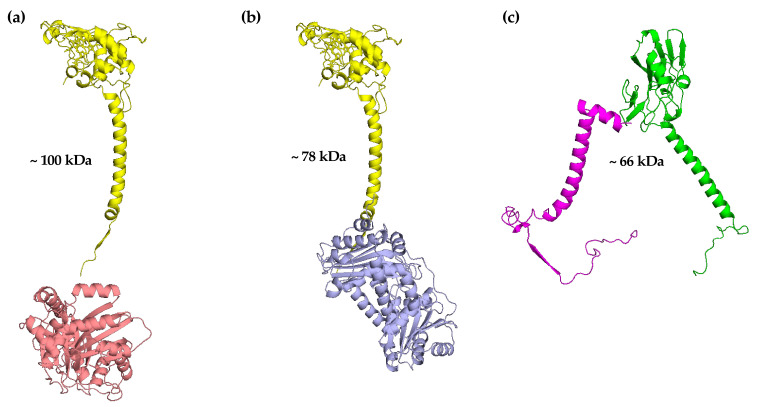
Additional *bc*_1_ sub-complexes. (**a**) Sub-complex containing cytochrome *c*_1_ (yellow) and core protein 2 (salmon). (**b**) Sub-complex containing cytochrome *c*_1_ and core protein 1 (light blue). (**c**) Sub-complex containing ISP (green) and Qcr9p (magenta). This figure shows the interaction between the indicated *bc*_1_ subunits on the basis of the crystal structure analysis carried out in yeast mitochondria.

**Table 1 ijms-23-10537-t001:** Protein subunits of the *bc*_1_ complex in *S. cerevisiae*.

Subunit	Amino Acids	Mr (kDa) *	Function	Prosthetic Group	Location	Structure
Cytochrome *b*	385	32	Catalytic	heme *b*_L_heme *b*_H_	Inner membrane	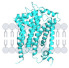
Cytochrome *c*_1_	248(309)	29	Catalytic	heme *c*_1_	Intermembrane space	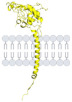
ISP	185(215)	22	Catalytic	2Fe-2S	Intermembrane space	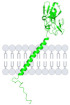
Core protein 1	431(457)	44	Non-redox	-	Matrix	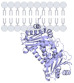
Core protein 2	352(368)	40	Non-redox	-	Matrix	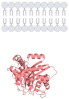
Qcr6p	122(147)	17	Non-redox	-	Intermembrane space	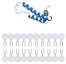
Qcr7p	126	14	Non-redox	-	Inner membrane/Matrix	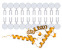
Qcr8p	93	11	Non-redox	-	Inner membrane	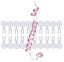
Qxr9p	65	7.3	Non-redox	-	Inner membrane	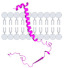
Qcr10p	76	8.5	Non-redox	-	?	

* Mr: apparent molecular mass obtained by SDS–PAGE analysis. Brackets indicate that the peptide is synthesised as a precursor protein. Interactions between subunits are described in the text.

## Data Availability

Not applicable.

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
