# Peer review of "Assembly of the Multi-Subunit Cytochrome bc1 Complex in the Yeast Saccharomyces cerevisiae"

_ijms, 2022, doi:10.3390/ijms231810537_

Round 1
Reviewer 1 Report
The manuscript entitled " Assembly of the multi-subunit cytochrome bc1 complex in the yeast Saccharomyces cerevisiae" is focused on the description of the molecular machinery described to be associated with the assembly of the cytochrome bc1 complex.
The manuscript is well written and the central theme fits perfectly within the scope of the International Journal of Molecular Sciences. Nevertheless, some minor improvements should be performed for its publication.
Minor considerations:
- Authors should present a table summarizing the name, function, localization, interaction and other relevant aspects of each subunit that compose the cytochrome bc1 complex.
- Concerning the full name Saccharomyces cerevisiae, usually only in the first time that the genus is used, it is written in full. In subsequent uses, the genus is abbreviated using the first initial and a period. This aspect should be taken into consideration in the manuscript.
- Another aspect to be considered in the genes and protein names. Authors should have into consideration the guidelines for formatting gene and protein names.
Author Response
Response to Reviewer 1 Comments
The manuscript entitled “Assembly of the multi-subunit cytochrome bc1 complex in the yeast Saccharomyces cerevisiae” is focused on the description of the molecular machinery described to be associated with the assembly of the cytochrome bc1complex.
The manuscript is well written and the central theme fits perfectly within the scope of the International Journal of Molecular Sciences. Nevertheless, some minor improvements should be performed for its publication.
We thank the Reviewer 1 for his comments.
Red text in the manuscript file indicates changes made in response to the suggestions of Reviewer.
Minor considerations:
Point 1: Authors should present a table summarizing the name, function, localization, interaction and other relevant aspects of each subunit that compose the cytochrome bc1 complex
Response 1: Table 1 has been added in the introduction section (pag. 2).
Point 2: Concerning the full name Saccharomyces cerevisiae, usually only in the first time that the genus is used, it is written in full. In subsequent uses, the genus is abbreviated using the first initial and a period. This aspect should be taken into consideration in the manuscript.
Response 2: We modified the text according to the Reviewer suggestion.
Point 3: Another aspect to be considered in the genes and protein names. Authors should have into consideration the guidelines for formatting gene and protein names.
Response 3: We modified the text according to the Reviewer suggestion (see also Reviewer 2 response).

Reviewer 2 Report
Dear Authors,
the manuscript presented for review is very well written in terms of its content. I believe that the content has been properly selected and grouped. The layout is clear and legible. However, I have some comments, which I will present below.
The first and most important remark concerns the correct writing of the names of genes, proteins and deletion mutants. Yeast genes are written in capital letters and italics, eg BCS1, the correct genotype of the shortened deletion mutant is (delta)bcs1 (bcs1 italic) or bcs1(delta) - the name of the mutant gene is written in lowercase and italic. If the authors write about expression, for example, "The expression of Bcs1p", the gene is expressed, so the correct spelling is "The expression of BCS1". I also suggest that the description should be standardized - if the authors of the paper use the notation eg Mzm1p for a protein, they should use this notation everywhere. In the manuscript, I noticed that the authors sometimes enter "p" for the protein name and sometimes not. I would like to request that the names of genes and proteins mentioned above be corrected throughout the manuscript. My second point is actually a suggestion. In my opinion, one more figure could be added to the manuscript - collective, summarizing. I think it could be a figure showing the location of cytochrome bc1 as part of the respiratory chain of MT.
Author Response
Response to Reviewer 2 Comments
Dear Authors,
the manuscript presented for review is very well written in terms of its content. I believe that the content has been properly selected and grouped. The layout is clear and legible.
However, I have some comments, which I will present below.
We thank the Reviewer 2 for his comments.
Red text in the manuscript file indicates changes made in response to the suggestions of Reviewer.
Point 1: The first and most important remark concerns the correct writing of the names of genes, proteins and deletion mutants. Yeast genes are written in capital letters and italics, eg BCS1, the correct genotype of the shortened deletion mutant is (delta)bcs1 (bcs1 italic) or bcs1(delta) - the name of the mutant gene is written in lowercase and italic. If the authors write about expression, for example, "The expression of Bcs1p", the gene is expressed, so the correct spelling is "The expression of BCS1". I also suggest that the description should be standardized - if the authors of the paper use the notation eg Mzm1p for a protein, they should use this notation everywhere. In the manuscript, I noticed that the authors sometimes enter "p" for the protein name and sometimes not. I would like to request that the names of genes and proteins mentioned above be corrected throughout the manuscript.
Response 1: We modified the text according to the Reviewer suggestion.
Point 2: My second point is actually a suggestion. In my opinion, one more figure could be added to the manuscript - collective, summarizing. I think it could be a figure showing the location of cytochrome bc1 as part of the respiratory chain of MT.
Response 2: According to the Reviewer suggestion, we added a new figure (Figure 1).

Reviewer 3 Report
This is a very well written review about bc1 complex assembly in yeast.
I have only one minor comment: In the introduction there was a remark regarding the unclear functional role of the non-catalytic subunits. This point could be a bit better elaborated in the part conclusions and perspectives.
Author Response
Response to Reviewer 3 Comments
This is a very well written review about bc1 complex assembly in yeast.
We thank the Reviewer 2 for his comments.
Red text in the manuscript file indicates changes made in response to the suggestions of Reviewer.
Point 1: I have only one minor comment: In the introduction there was a remark regarding the unclear functional role of the non-catalytic subunits. This point could be a bit better elaborated in the part conclusions and perspectives.
Response 1: As suggested by the Reviewer, we discussed this aspect in the “Conclusions and perspectives” section (pag. 13).
